# Mineralogical Characterization of Lithomargic Clay Deposits along the Coastal Belt of Udupi Region of South India

Deepak Nayak , Purushotham G. Sarvade *, Udaya Shankara H. N. and Jagadeesha B. Pai *

Department of Civil Engineering, Manipal Institute of Technology, Manipal Academy of Higher Education, Manipal 576104, Karnataka, India
* Correspondence: pg.sarvade@manipal.edu (P.G.S.); jaga.pai@manipal.edu (J.B.P.)

**Abstract:** Lithomargic clay is generally found below the lateritic soil along the coastal belt of Karnataka. It is rich in silt content and dispersive in nature. This type of soil is liable to erosion and landslides. The lithomargic clay is largely found in the western coast of South India. At present, coastal belt of Udupi district is witnessing a lot of developments in terms of industry, infrastructures, and other activities. Lithomargic clay is a type of problematic soil, which needs a thorough study to make it suitable to sustain any engineering structure such as buildings, pavements, railways, dams. A characterization and mineralogical study is conducted to identify the presence of minerals and compounds for the various soil samples collected along the coastal belt of Udupi regions using X-ray diffraction (XRD), energy dispersive X-ray spectroscopy (EDS), and scanning electron microscopy (SEM) analysis. The primary minerals observed in majority of the regions are quartz, feldspar such as orthoclase, muscovite, and the secondary minerals formed by the decomposition and chemical alteration of primary minerals include sheet minerals such as kaolinite, halloysite, dickite, gibbsite, and illite in high proportions. The study also shows the presence of iron compounds such as fayalite, goethite, and siderite. The majority of the elements observed are oxygen, silicates, aluminum, potassium, and iron which confirms the presence of the compounds identified through XRD analysis.

**Keywords:** lithomargic clay; mineralogy; characterization; XRD analysis; EDS analysis; SEM analysis

## 1. Introduction

The coastal belt of Karnataka has various types of soil with varying properties. The most commonly and abundantly available soils are lateritic soil and lithomargic clay. Lithomargic clays are deposited below lateritic formations at shallow depths, sandwiched between the parent granitic gneiss beneath and the hard lateritic crust above as shown in Figure 1 [1–3]. As most of the industries require good source of water, access though roadways, waterways, and airways, more and more industries and developments are progressing near coastal belts. As the soil properties changes from place to place, a rigorous study is required to identify the strength and weakness of the lithomargic clay. In this paper, a mineralogical characterization is carried out to identify the different elements and minerals present in the lithomargic clays of Udupi district.

The upper ferricretic layer is characterized by a sandy-gravelly grain size, whilst in the limonitic, transition, and saprolitic zones the silt-clay fraction predominates. Bolla et al. [4] discovered that the low friction angle of the clay layers and the high permeability of the limestone angular gravel were liable for the reduction in shear strength that corresponded with the basal rupture zone, favoring the enormous land sliding. Lithomargic clay, which is dispersive by nature and very vulnerable to erosions because of the heavy rainfall that alters the groundwater table, resulting in excessive settlements [3].

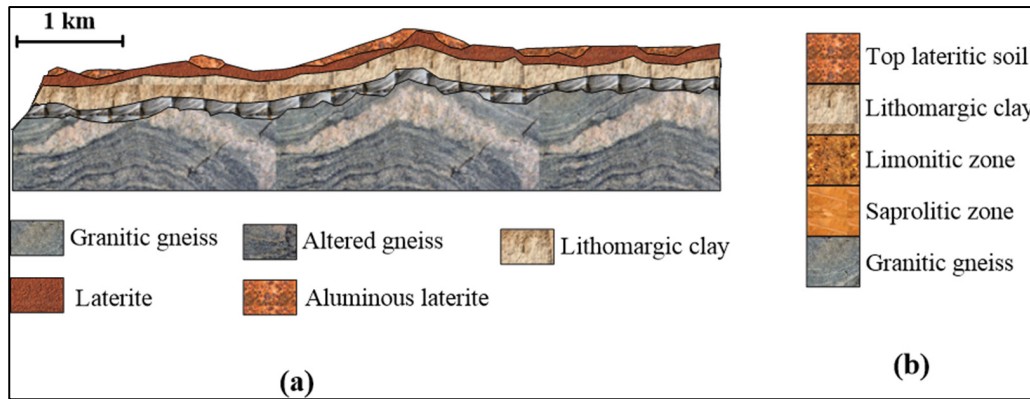

**Figure 1.** Soil stratification, (**a**) laterization on soil made up of gneissic granites (**b**) showing different zones of the laterite profile.

These dispersive soils are particularly prone to erosion [5]. This problematic residual soil called lithomargic clay, which is pinkish and whitish in color and primarily made up of illite and kaolinite, is found at a depth of 1 to 5 m below the lateritic outcrop. It is created as a result of tropical and subtropical weathering of laterite soils [6]. The soil chemistry, mineralogy, dissolved salts in the pore water, and eroding water are some of the variables that affect how easily clays disperse or de-flocculate. The experiment includes blending the soil with 2, 4, and 6% lime and concluded that the dispersion of lithomargic clay reduced to 12.9% with the 6% addition of lime [7]. A significant increase in the strength was observed with the inclusion of randomly distributed coir reinforcement [8]. The inclusion of coir embedded as mat reinforcement also resulted in improvement of strength of lithomargic clay [9].

Below the ferruginous zone, the primary forms of iron were goethite or hematite, while the main forms of aluminum were kaolinite or gibbsite. Based on their mineralogy, these deposits are divided into three categories: aluminous laterites, ferruginous laterites, and lithomargic clay. Laterites were created in situ by altering parent rocks with granite/granitic gneiss compositions to produce their major, trace, and mineralogical components. Major mineral changes that occur during the weathering of granite or granitic gneiss include the transformation of feldspar into montmorillonite, illite, kaolinite or halloysite, and ultimately bauxite. Despite the climate temperature, gibbsite, quartz, and kaolinite dominate the remnant lateritic profile [10]. The best results were obtained with lithomargic clay mixed with 25% granulated blast furnace slag (GBFS). With the addition of 2 and 4% cement to the optimized slag–soil mix, the strength was further increased. The increase in strength was discovered by SEM and XRD research, which also allowed for the observation of microstructural alterations. The increase in strength characteristics of lithomargic soil was attributed to calcium silicate hydrates (C-S-H), calcium aluminate silicate hydrates (C-A-S-H), calcium silicate hydroxide hydrate (CSHH), and a few other related compounds [11]. The XRD analysis of lithomargic clay showed the presence of minerals like Gibbsite, Kaolinite, Biotite, and Muscovite [12]. Laita et al. [13] have investigated the mineralogical and textural alterations in bauxite with illite- and kaolinite-rich clays burned between 1000 and 1270 °C, which certainly impact the materials' physical properties. Mullite is produced at 1000 degrees Celsius. The continuing instability of the first phases leads to the formation of a Si and Al-rich vitreous phase from which cristobalite, ilmenite, hercynite, mullite, and corundum crystalize at elevated temperature. The creation of vitreous phase and the crystallization of mullite are linked to increases in density and linear shrinkage, which are correlated with decreases in porosity, water absorption, and thermal conductivity. At high temperatures, mullite and corundum are the most prevalent phases, with mullite content being higher in samples containing illite and kaolinite [13]. Bhat and Nayak [14] conducted experiments to increase the strength of lithomargic clay by employing lime and granulated blast furnace slag to chemically stabilize the soil. Con-

sidering the disposal of vast volumes of industrial waste, known as GBFS, has become an environmental issue, using it for soil stabilization is a long-term solution. Laboratory studies were conducted on lithomargic clay, replacing it with variable amounts of GBFS and adding different percentages of lime, to optimize the usage of lime and explain the process behind the increase in strength. The optimal lime and GBFS content was discovered to be 4% and 20%, respectively. A considerable strength enhancement was obtained when lithomargic clay was mixed with 4% lime and 20% GBFS for further optimization. The increase in strength was determined by SEM and XRD examinations of the stabilized soil, which revealed structural alterations and the creation of compounds such as CASH, CSHH, CAOH, gyrolite, and gismondine [14]. Narloch et al. [15] analyzed the impact of soil mineral composition on cement stabilized rammed earth (CSRE) compressive strength. On CSRE samples with different mineral compositions but maintaining same particle size distribution, cement content, and water content, compression tests were conducted. It was discovered that based on the compression strength measurements and examined SEM images, the CSRE compressive strength was dramatically altered by variations in mineral composition. The results showed that the montmorillonite reduced the compressive strength substantially and also the beidellite to a lesser extent. Kaolinite slightly increased the compressive strength.

When GGBS and fly ash were used as replacement in the soil in proportions of 20%, 30%, and 40%, the unconfined compressive strength (UCS) and California bearing ratio (CBR) increased, and the weight-to-volume ratio of silica oxide to sodium oxide solution was kept constant at 1.25 with varying doses of 2, 3, and 4% sodium oxide [16]. When lithomargic clay was stabilized with cement, the XRD examination showed that ettringite, calcium silicate hydrate (CSH), and calcium aluminate hydrate (CAH) were formed. The SEM study revealed that the cement and quarry dust additions had altered the soil's structure [17].

As the minerals present in the soil are responsible for the physical and chemical behavior of soil, it is very important to know the minerals present and its variation along the study region. By knowing the mineral characteristics and its behavior, one can decide on the suitable ground improvement techniques to be applied on soil to overcome the shortcomings in soil. In the present study, characterization and mineralogical properties are evaluated to identify the presence of minerals and compounds for the various soil samples collected along the coastal belt of Udupi regions using X-ray diffraction (XRD), energy dispersive X-ray spectroscopy (EDS), and scanning electron microscopy (SEM) analysis to discuss and find the correlation of mineralogy with geotechnical parameters and suggest the suitable ground improvement techniques to be applied on soil.

## 2. Materials and Methods

### 2.1. Lithomargic Clay

The study consists of collecting the soil samples from the coastal belt of Udupi region of Karnataka state in southern parts of India. Soil samples were collected from various places along the coastal belt of Udupi region namely Katapadi, Alevoor, Manipal, Kolalgiri, Brahmavar, Kumbashi, Hemmadi, Valandhur, Nagoor, and Byndoor.

The Figure 2a–c shows the soil sample collected from the location Katpadi, Alevoor, and Manipal. Figure 2d–f shows the samples collected from Kolalgiri, Brahmavar, and Kumbashi respectively. Figure 2g–j shows the samples collected from Hemmadi, Valandhur, Nagoor, and Byndoor respectively.

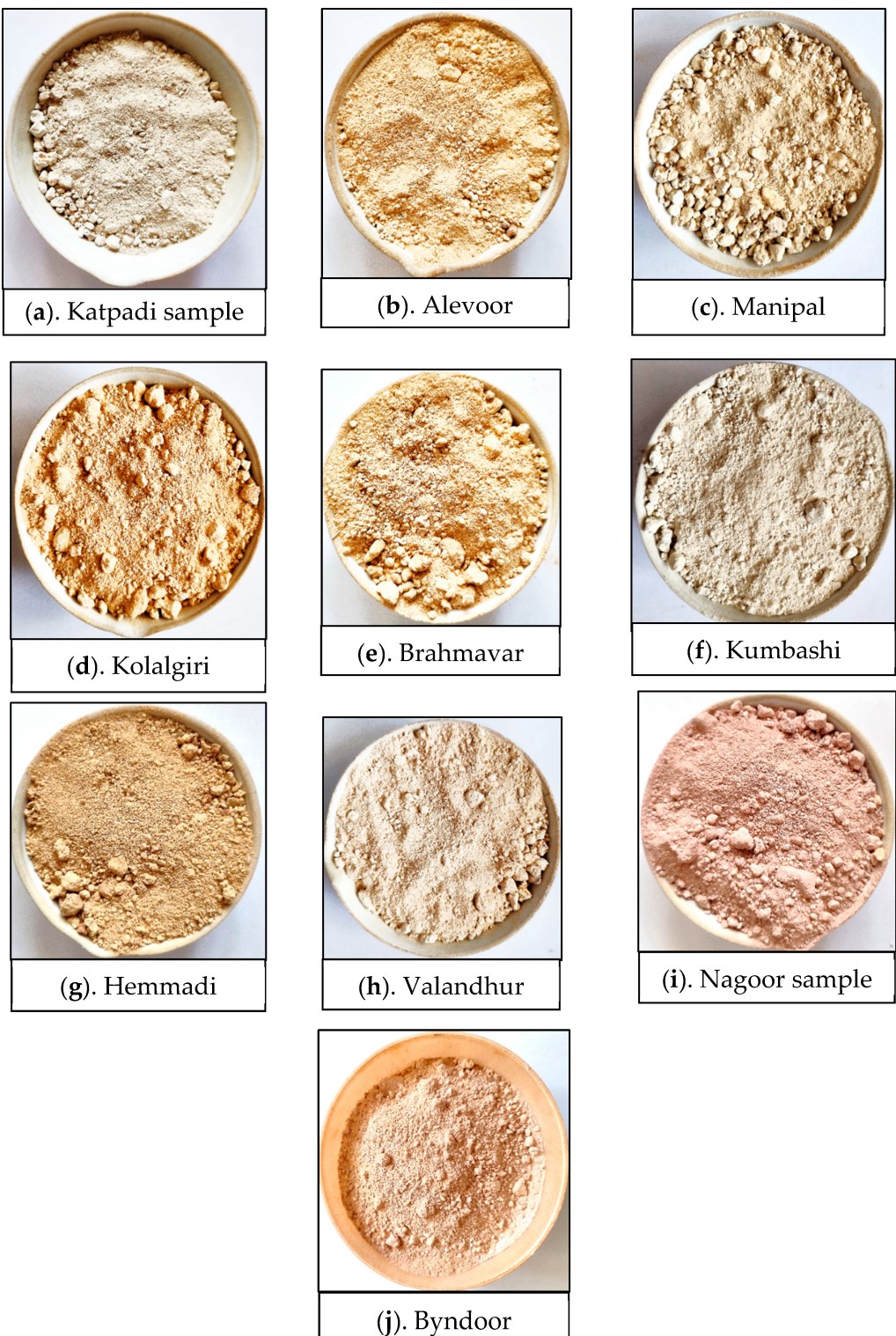

**Figure 2.** Samples from various locations in Udupi region.

*2.2. Characterization of Lithomargic Clay along the Coastal Belt of Udupi District*

2.2.1. Mineralogical Tests on Collected Samples

Mineralogical analysis include scanning electron microscopy (SEM) and energy dispersive X-ray spectroscopy (EDS) and are used to determine the mineralogical composition of the soil samples. The oven dried soil samples passing through 75 μ sieve are taken for the conduction of mineralogical tests. Prior to the test, gold sputtering must be carried

out over the surface of the soil. The obtained results are analyzed to identify the elements present in the soil.

### 2.2.2. Scanning Electron Microscopy (SEM) and Energy Dispersive X-ray Spectroscopy (EDS)

Soil passing through 75 μ sieve is collected and gold sputtering is carried out on the sample. Scanning electron microscopy is conducted on the sample in order to understand the microstructure of the soil sample. The EDS is used to analyze the chemical compositions of the specimen. Energy dispersive X-ray spectroscopy is performed to study the elements present in the soil and to quantify the major elements present in the soil.

### 2.2.3. X-ray Diffraction (XRD) Analysis

Soil passing through 425 μ sieve is collected for the XRD analysis. XRD analysis is performed to identify the minerals present in the soil samples.

## 3. Results and Discussion

The soil samples were collected from ten different places namely Katapadi, Alevoor, Manipal, Kolalgiri, Brahmavar, Kumbashi, Hemmadi, Vallandhur, Nagoor, and Byndoor.

*Mineralogical Tests*

The Micro fabric and mineralogical tests performed on the soil samples are scanning electron microscopy (SEM) which gives the microscopic image of the soil which is used to analyze the structure of the elements, minerals present in the soil. The energy dispersive X-ray spectroscopy (EDS) is used to determine the composition of the soil samples. The results of the micro fabric and mineralogical tests performed on the soil are presented below. The elements that were identified in the soil samples are C, O, Al, Si, K, Fe, Cr. The percentage of the various elements identified are also obtained in this test. The percentage of elements in terms of weight and corresponding atomic percentage of same elements are obtained in the same scan area. The results presented below shows the SEM images of the soil samples, EDS spectra of the soil samples, and proportions of various elements found in the soil samples obtained from the EDS test.

The EDS analysis of Katpadi samples Figure 3a,b shows the elements weight percentage and atomic weight percentage respectively, whereas Figure 3c,d shows the SEM image and spectrum image respectively for the samples collected from Katpadi location.

Figure 4a,b shows the minerals and compounds identified through XRD analysis and their quantification respectively for the samples collected from Katpadi location.

As per the quantification results (Figure 4b), it can be observed that the Katpadi soil sample consists of relatively higher proportions of dickite ($Si_8Al_{18}O_{36}H_{16}$) (18%), gibbsite ($Al_8O_{24}H_{24}$) (18%), kaolinite ($Al_2Si_2O_9H_4$) (15%), tridymite ($Si_{80}O_{160}$) (15%), and quartz ($Si_6O_6$) (10%). Fayalite ($Fe_8Si_4O_{16}$) (8%), siderite ($Fe_6C_6O_{18}$) (4%), eskolaite ($Cr_{12}O_{18}$) (2%), iron silicates (7%), and cristobalite (3%) are present at very low proportions. The SEM image from Figure 3c shows sheet minerals such as dickite, kaolinite, and eskolaite as witnessed by the EDS analysis showing the presence of aluminum, silica, and chromium.

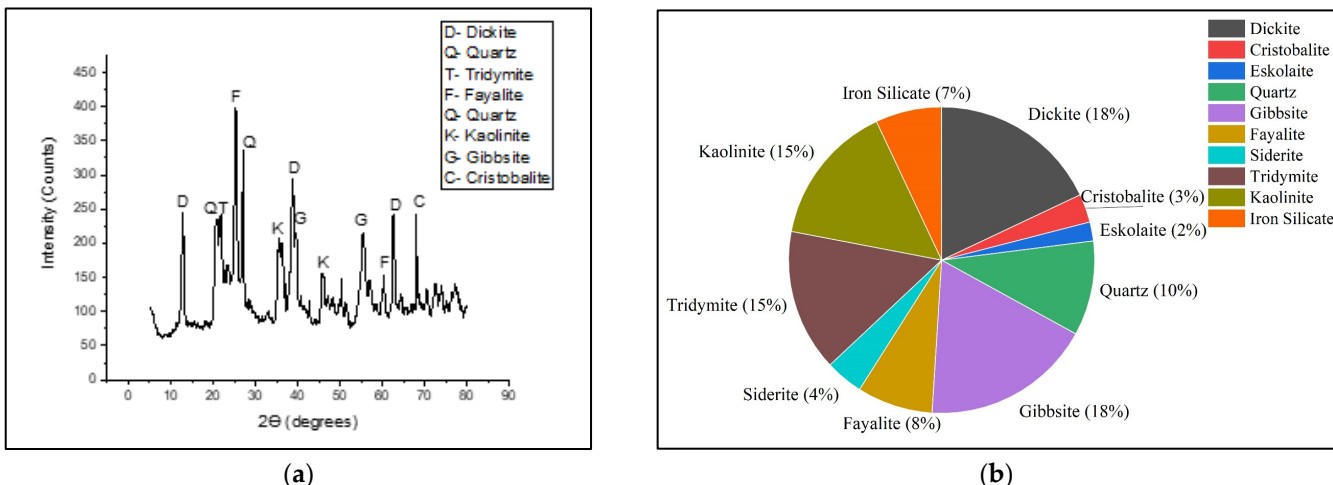

**Figure 3.** EDS analysis of Katpadi sample, (**a**) elements weight (%), (**b**) atomic weight (%), (**c**) SEM image, (**d**) spectrum image.

**Figure 4.** Characterization of minerals and compounds, (**a**) XRD analysis of Katapadi sample, (**b**) quantification of compounds.

The compounds observed in the XRD analysis (Figure 4a) are evident from the elements observed by the EDS analysis as shown in Figure 3a,d. The majority of the elements observed are oxygen, silicates, and aluminum followed by carbon and chromium. In this

sample, the aluminum and silicate minerals such as dickite, kaolinite, and gibbsite can be found in large proportions. Ferrous compounds are found at low proportions. The presence of elements such as Fe, Al, and Si also confirms the presence of iron silicates and fayalite. The quartz, dickite, kaolinite, and fayalite minerals can be observed with high intensity in the XRD analysis shown in Figure 4a.

Mineralogical analysis of Alevoor samples are shown in Figures 5 and 6. The EDS analysis of Alevoor sample as in Figure 5a,b shows the identified elements weight percentage and atomic weight percentage respectively, whereas Figure 5c,d show the SEM image and spectrum image of Alevoor sample respectively.

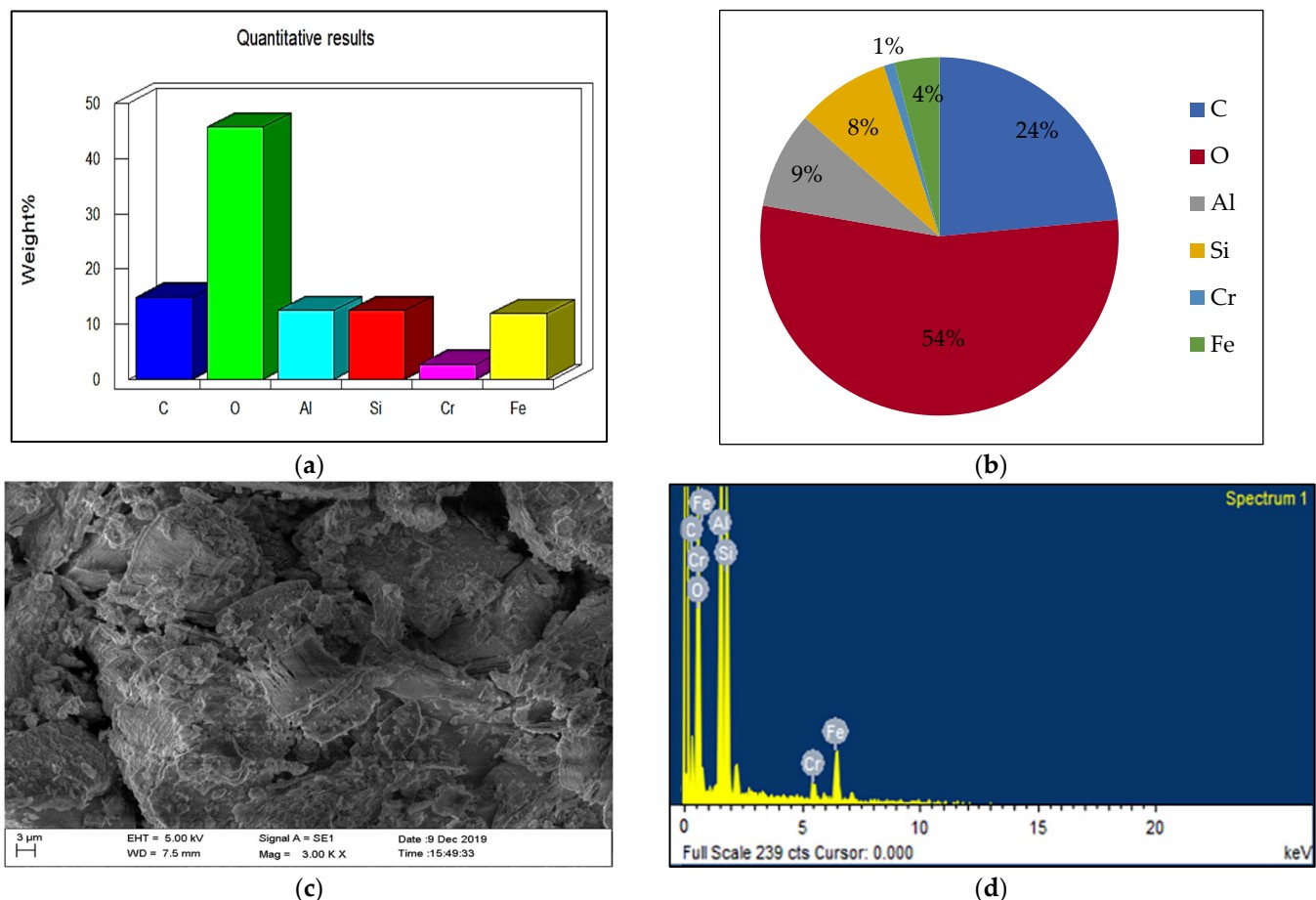

**Figure 5.** EDS analysis of Alevoor sample; (**a**) elements weight (%), (**b**) atomic weight (%), (**c**) SEM image, (**d**) spectrum image.

Figure 6a,b shows the minerals and compounds identified through XRD analysis and their quantification respectively for the samples collected from Alevoor location.

The SEM image in Figure 5c shows the presence of sheet minerals such as kaolinite and closely packed large-sized booklets identified as nacrite. The Figure 5a,b shows the elements and the weight percentage of the Alevoor sample. The elements such as aluminum, silica, iron, chromium, carbon, and oxygen are evident from the EDS analysis.

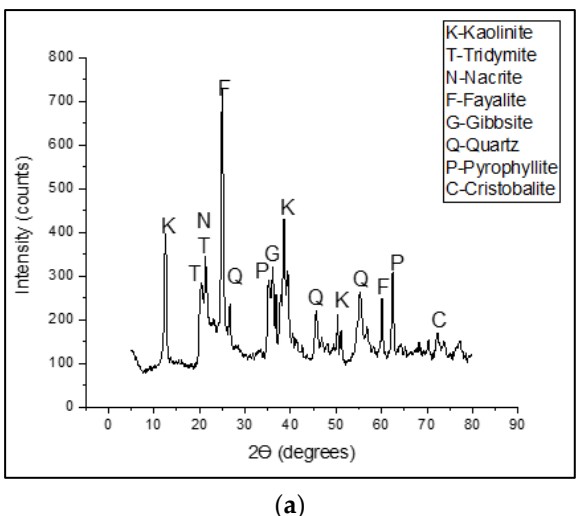

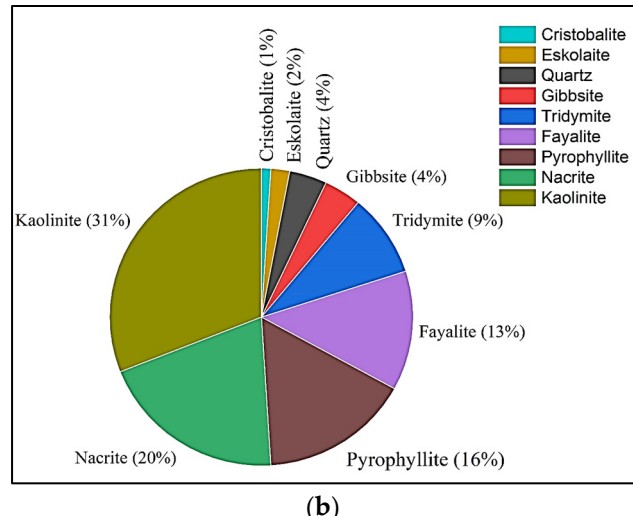

(**a**)

(**b**)

**Figure 6.** Characterization of minerals and compounds; (**a**) XRD analysis of Alevoor sample; (**b**) quantification of compounds.

The Figure 6a shows the XRD analysis of Alevoor sample. From the XRD analysis, the minerals observed are kaolinite, nacrite, fayalite, gibbsite, pyrophyllite, and tridymite which have high intensity peaks. The quantification of minerals are as shown in Figure 6b. As per quantification of results, it can be observed that the soil sample consists of relatively higher proportions of kaolinite (31%), nacrite (20%), pyrophyllite ($Al_2Si_4O_{12}$) (16%), and fayalite (13%). The quartz (4%), tridymite (9%), and eskolaite (2%) are present at very low proportions. The compounds observed in the XRD analysis are evident from the elements observed by the EDS analysis as shown in Figure 5a,b. The majority of the elements observed are oxygen, silicates, and aluminum, carbon followed by iron and chromium. Hydrogen cannot be detected by EDS test.

Mineralogical analysis of Manipal samples are shown in Figures 7 and 8. The EDS analysis of Alevoor sample as in Figure 7a,b shows the identified elements weight percentage and atomic weight percentage respectively, whereas Figure 7c,d shows the SEM image and spectrum image of the Manipal sample, respectively.

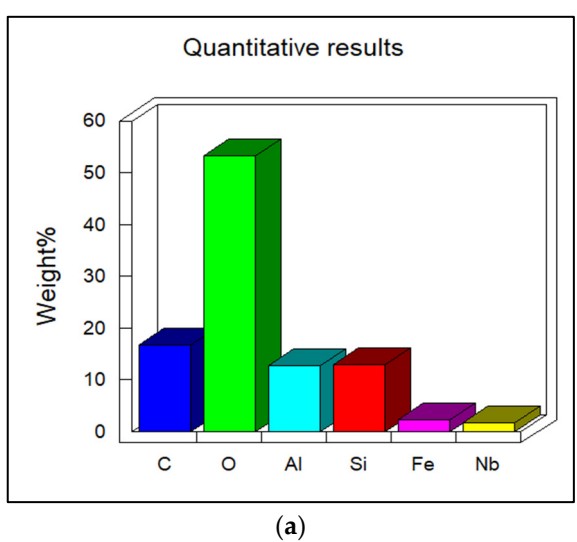

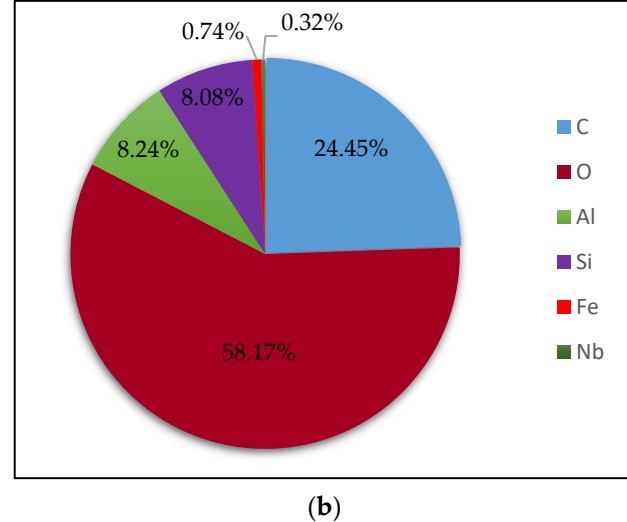

(**a**)

(**b**)

**Figure 7.** *Cont.*

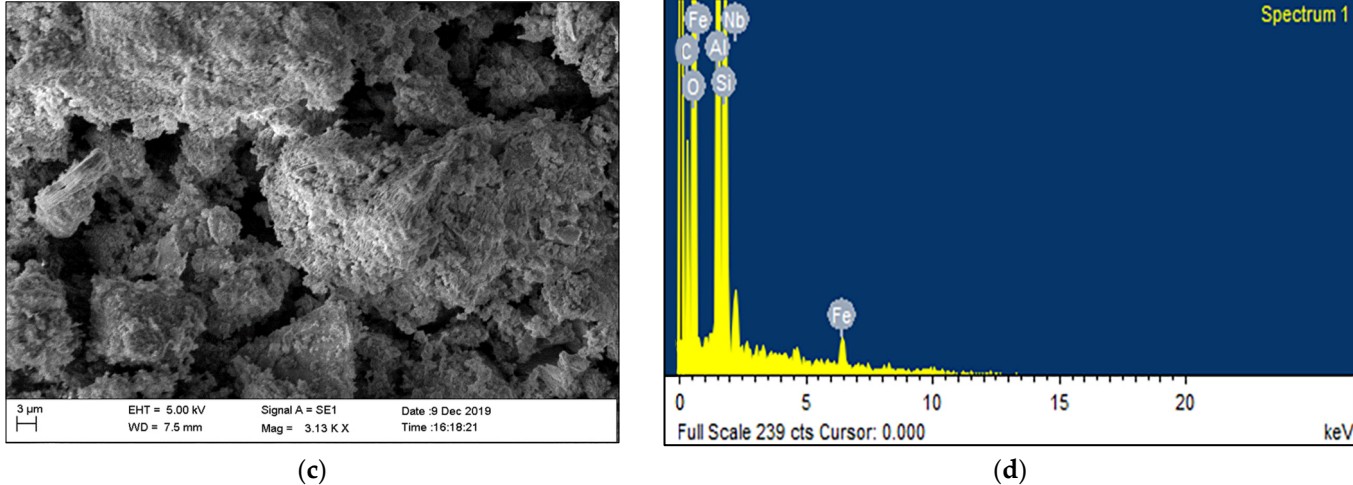

**Figure 7.** EDS analysis of Manipal sample; (**a**) elements weight (%), (**b**) atomic weight (%), (**c**) SEM image, (**d**) spectrum image.

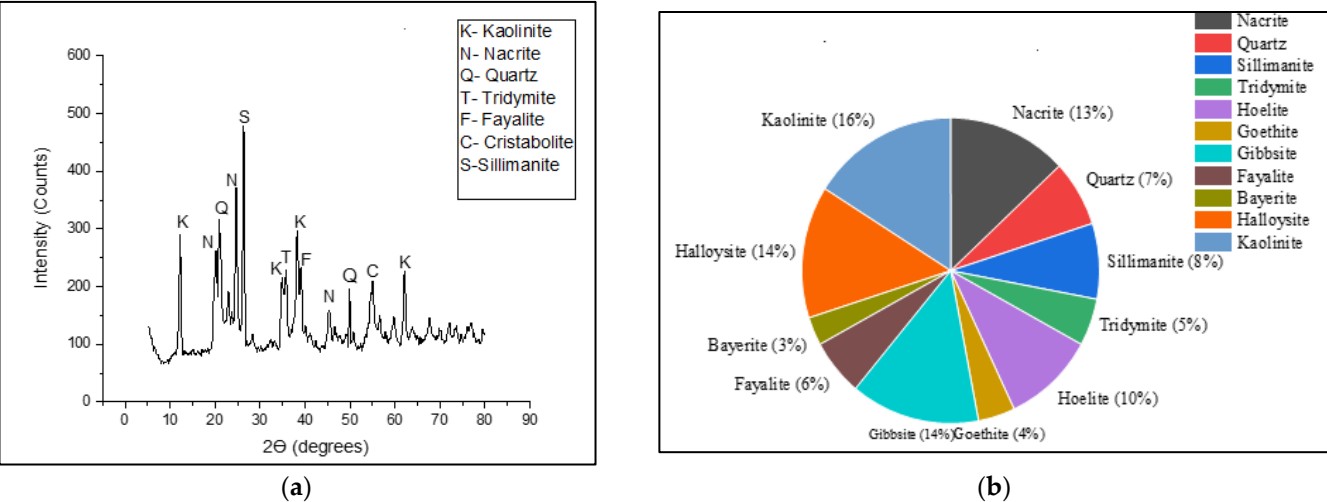

**Figure 8.** Characterization of minerals and compounds; (**a**) XRD analysis of Manipal sample; (**b**) quantification of compounds.

Figure 8a,b shows the minerals and compounds identified through XRD analysis and their quantification respectively for the samples collected from Manipal location.

The quantification of minerals is shown in Figure 8b. As per quantification results, it can be observed that the soil sample consists of relatively higher proportions of kaolinite (16%), halloysite (14%), nacrite (13%) hoelite (10%), and sillimanite (8%). Quartz (7%), bayerite (3%), tridymite (5%), and goethite (4%) are present at very low proportions. The halloysite is not showing a proper peak intensity but may be associated with low intensity peaks, constituting 14% of the total volume. The SEM image (Figure 7c) shows the presence of fayalite having irregular shaped poorly defined crystallographic facets. The compounds observed in the XRD analysis (Figure 8a) are evident from the elements observed by the EDS analysis shown in Figure 7a,d. The majority of the elements observed are oxygen, silicates, aluminum followed by carbon, iron, and niobium.

Mineralogical analysis of Kolalgiri samples are shown in Figures 9 and 10. The EDS analysis of Kolalgiri sample as in Figure 9a,b shows the identified elements weight percentage and atomic weight percentage, respectively, whereas Figure 9c,d shows the SEM image and spectrum image of Kolalgiri sample respectively.

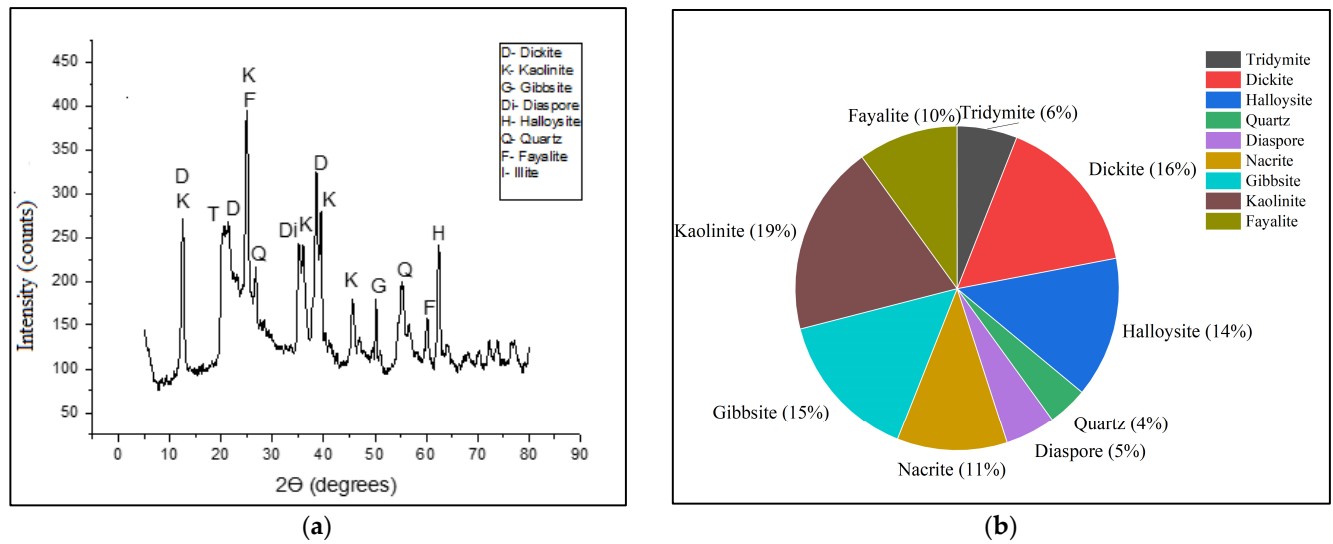

**Figure 9.** EDS analysis of Kolalgiri sample; (**a**) elements weight (%), (**b**) atomic weight (%), (**c**) SEM image, (**d**) spectrum image.

**Figure 10.** Characterization of minerals and compounds; (**a**) XRD analysis of Kolalgiri sample, (**b**) quantification of compounds.

Figure 10a,b shows the minerals and compounds identified through XRD analysis and their quantification respectively for the samples collected from Kolalgiri location.

The quantification of minerals (Figure 10b) depicts that the soil sample consists of relatively higher proportions of kaolinite (19%), dickite (16%), gibbsite (15%), halloysite (14%), nacrite (11%), and fayalite (10%). Quartz (4%) and diaspore (5%) are present at very low proportions. Kaolinite, fayalite, gibbsite, and halloysite compounds show a proper peak intensity in XRD analysis (Figure 10a). The compounds observed in the XRD analysis are evident from the elements observed by the EDS analysis as shown in Figure 9a,d. The majority of the elements observed are oxygen, silicates, and aluminum followed by iron and carbon. In this sample, the alumina- and silicate-based compounds such as kaolinite, dickite, gibbsite, and halloysite can be found in large proportions. The EDS analysis shows the presence of iron (Fe) and confirms the presence of fayalite. The mineral depicted by the SEM image analysis (Figure 9c) can be identified as kaolinite and gibbsite considering the elemental analysis and layered structure.

Mineralogical analysis of Brahmavar samples are shown in Figures 11 and 12. The EDS analysis of Brahmavar sample as in Figure 11a,b shows the identified elements weight percentage and atomic weight percentage respectively whereas Figure 11c,d shows the SEM image and spectrum image of Brahmavar sample respectively.

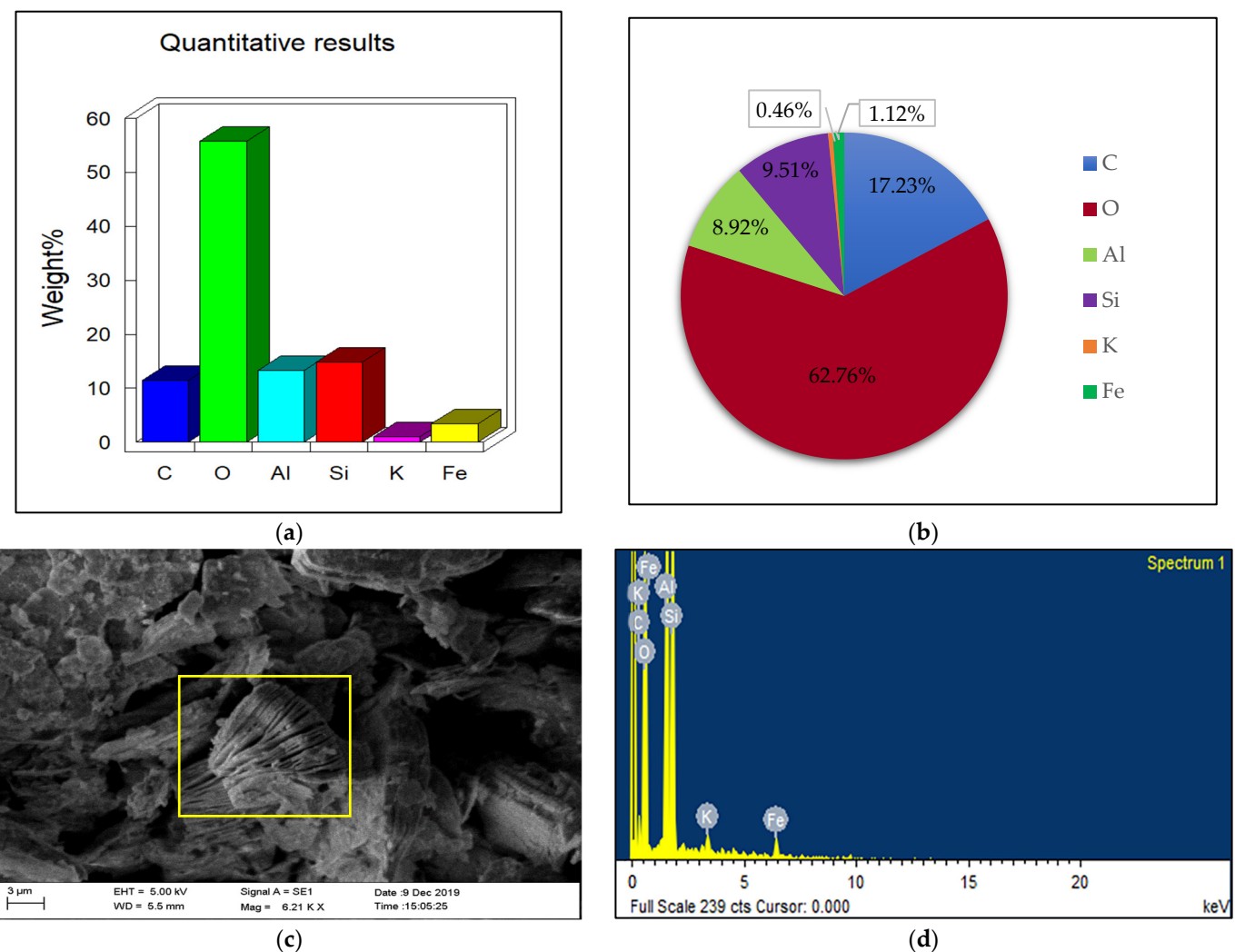

**Figure 11.** EDS analysis of Brahmavar sample; (**a**) elements weight (%), (**b**) atomic weight (%), (**c**) SEM image, (**d**) spectrum image.

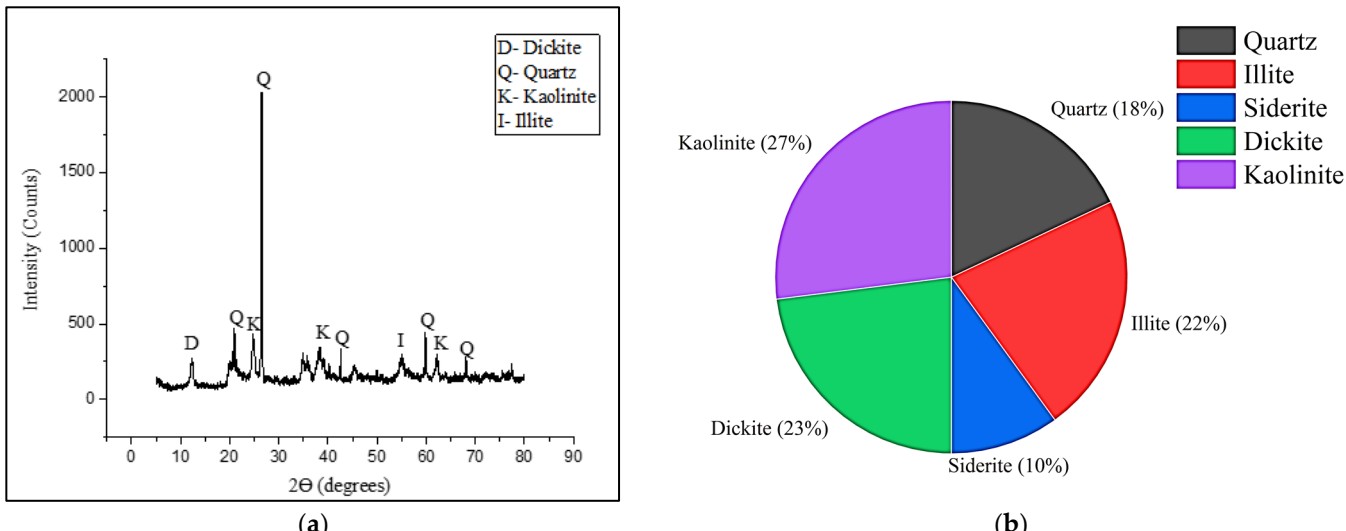

(a)                                         (b)

**Figure 12.** Characterization of minerals and compounds. (**a**) XRD analysis of Brahmavar sample, (**b**) quantification of compounds.

Figure 12a,b shows the minerals and compounds identified through XRD analysis and their quantification respectively for the samples collected from Brahmavar location.

The quantification of minerals are as shown in Figure 12b. As per quantification results, it can be observed that the soil sample consists of relatively higher proportions of kaolinite (27%), dickite (22%), illite (22%), and quartz (18%). Siderite (10%) is present at low proportion. The quartz, dickite, illite, and kaolinite compounds show a proper peak intensity in XRD analysis (Figure 12a). The minerals of SEM image as shown in Figure 11c can be identified as kaolinite with layered sheets of booklet.

The compounds observed in the XRD analysis are evident from the elements observed by the EDS analysis as shown in Figure 11a,b,d. The majority of the elements observed are oxygen, silicates, aluminum, potassium, and iron. In this sample, potassium-, alumina-, silicate-, and iron-based compounds such as kaolinite, illite, dickite, and quartz are found in large proportions. The iron (Fe) and carbon (C) also witness the presence of siderite.

Mineralogical analysis of Kumbashi samples are shown in Figures 13 and 14. The EDS analysis of Kumbashi sample as in Figure 13a,b shows the identified elements weight percentage and atomic weight percentage respectively, whereas Figure 13c,d shows the SEM image and spectrum image of Kumbashi sample respectively.

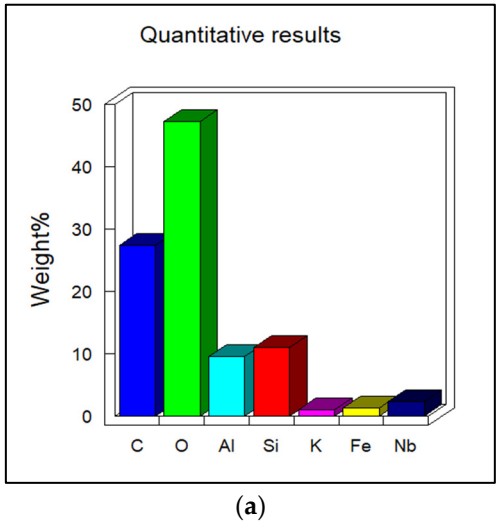
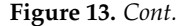

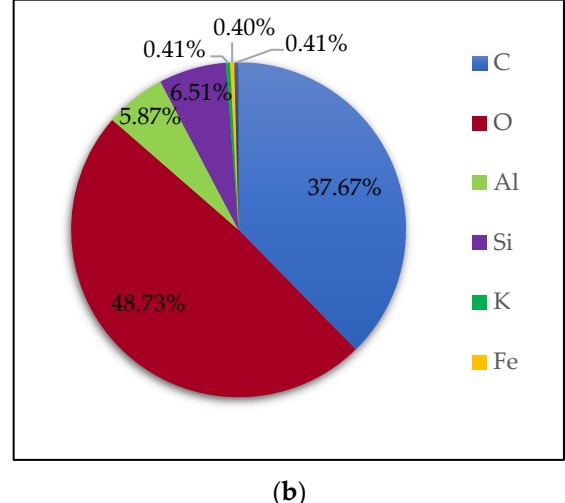

(a)                                         (b)

**Figure 13.** *Cont.*

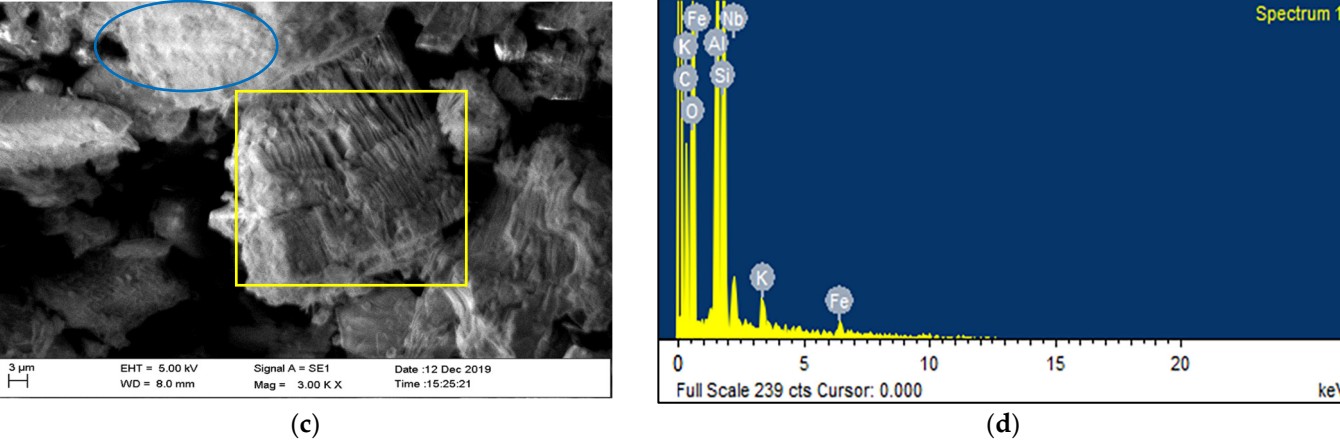

**Figure 13.** EDS analysis of Kumbashi sample; (**a**) elements weight (%), (**b**) atomic weight (%), (**c**) SEM image, (**d**) spectrum image.

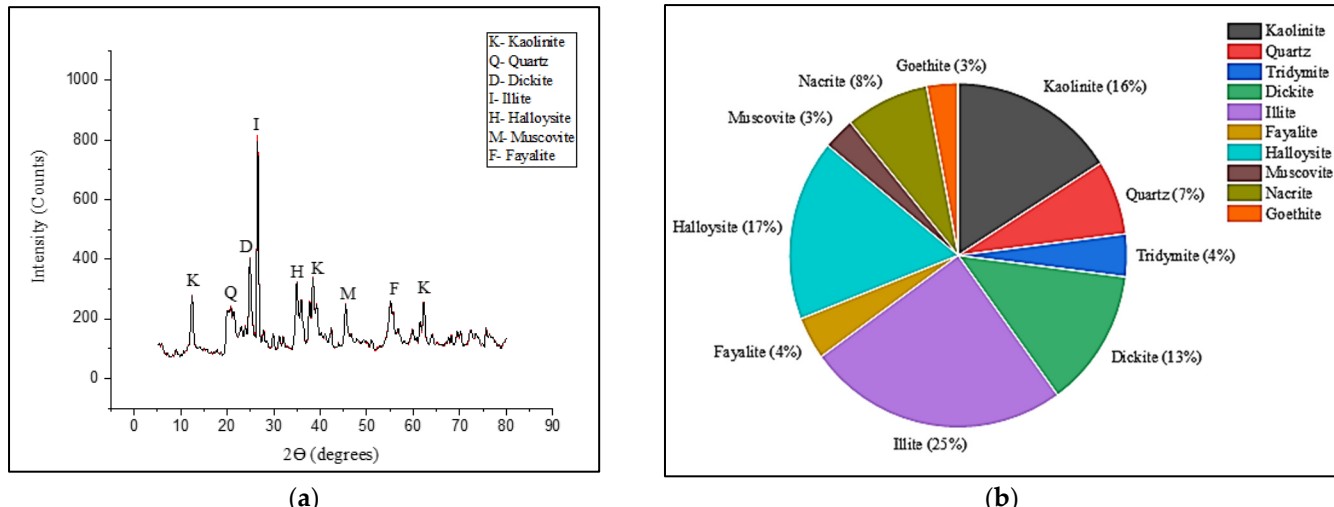

**Figure 14.** Characterization of minerals and compounds; (**a**) XRD analysis of Kumbashi sample, (**b**) quantification of compounds.

Figure 14a,b shows the minerals and compounds identified through XRD analysis and their quantification respectively for the samples collected from Kumbashi location.

From the SEM image as in Figure 13c, the compound can be identified as kaolinite having layered sheet mineral and illite with sponge-like structure. The quantification of minerals is shown in Figure 14b. As per quantification of results, it can be observed that the soil sample consists of relatively higher proportions of illite (13.9%), halloysite (17%) and dickite (13%). Muscovite (3%), goethite (3%), and tridymite (4%) are present at very low proportions.

The compounds observed in the XRD analysis (Figure 13a) are evident from the elements observed by the EDS analysis as shown in Figure 13d. The majority of the elements observed are oxygen, silicates, and aluminum followed by potassium and iron. In this sample, the potassium-based compounds such as muscovite and illite can be commonly observed. The presence of both elements Fe and K also confirms the presence of illite and goethite. The quartz, kaolinite, muscovite, and halloysite minerals can be observed with high intensity in the XRD analysis (Figure 14a).

Mineralogical analysis of Hemmadi samples are shown in Figures 15 and 16. The EDS analysis of Hemmadi sample as in Figure 15a,b shows the identified elements weight percentage and atomic weight percentage respectively; whereas Figure 15c,d shows the SEM image and spectrum image of Hemmadi sample respectively.

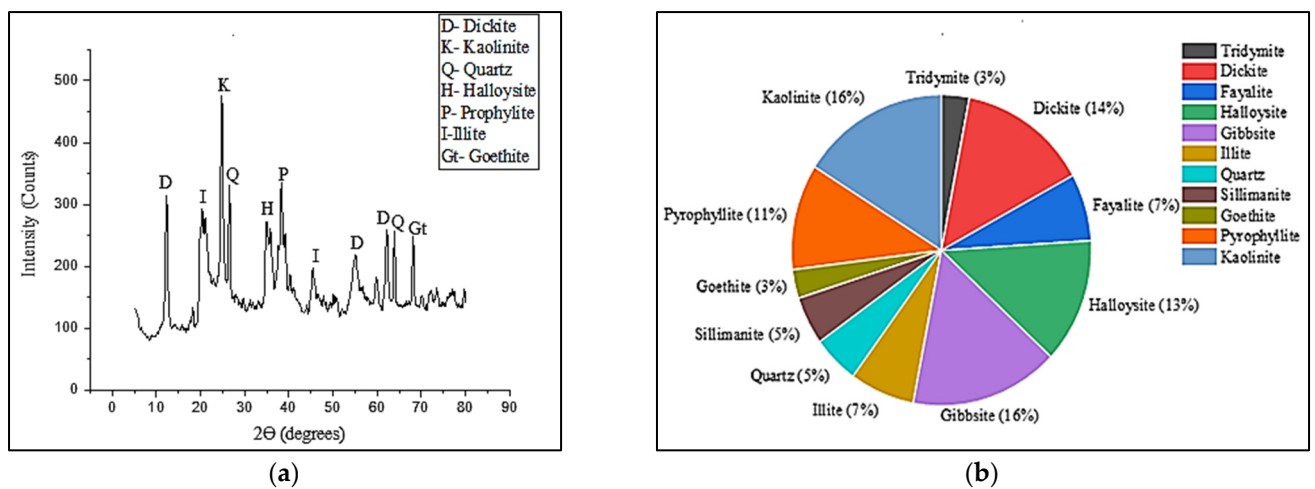

**Figure 15.** EDS analysis of Hemmadi sample; (**a**) elements weight (%), (**b**) atomic weight (%), (**c**) SEM image, (**d**) spectrum image.

**Figure 16.** Characterization of minerals and compounds; (**a**) XRD analysis of Hemmadi sample, (**b**) quantification of compounds.

Figure 16a,b shows the minerals and compounds identified through XRD analysis and their quantification respectively for the samples collected from Hemmadi location.

The quantification of minerals are as shown in Figure 16b. As per quantification of results, it can be observed that the soil sample consists of relatively higher proportions of dickite (14%), gibbsite (16%), and also signifies the presence of halloysite (13%), pyrophyllite (11%). Sillimanite (5%) and tridymite (3%) are present at very low proportions. The SEM image of Figure 15c shows the presence of fibrous structure such as illite.

The compounds observed in the XRD analysis are evident from the elements observed by the EDS analysis as shown in Figure 15a,d. The majority of the elements observed are oxygen, carbon, silicates, and aluminum followed by potassium and iron.

The presence of kaolinite, dickite, halloysite, goethite, and quartz compounds can be observed in the XRD analysis (Figure 16a) as it depicts a clear peak intensity. The main elements of these compounds such as aluminum, silica, potassium, and iron are evident from the EDS analysis as shown in Figure 15a,d.

Mineralogical analysis of Valandhur samples are shown in Figures 17 and 18. The EDS analysis of Valandhur sample as in Figure 17a,b shows the identified elements weight percentage and atomic weight percentage respectively, whereas Figure 17c,d show the SEM image and spectrum image of Valandhur sample, respectively.

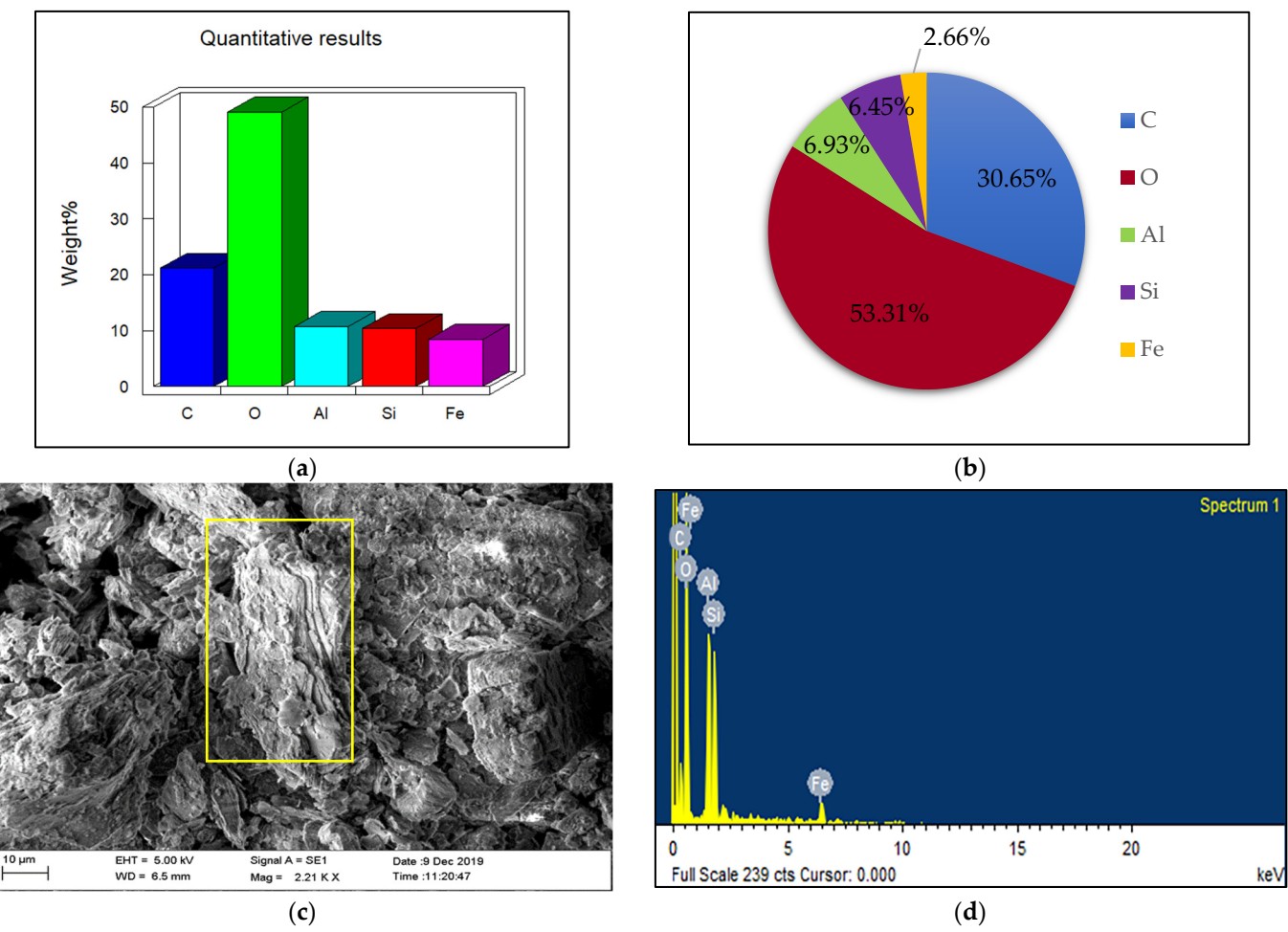

**Figure 17.** EDS analysis of Valandhur sample; (**a**) elements weight (%), (**b**) atomic weight (%), (**c**) SEM image, (**d**) spectrum image.

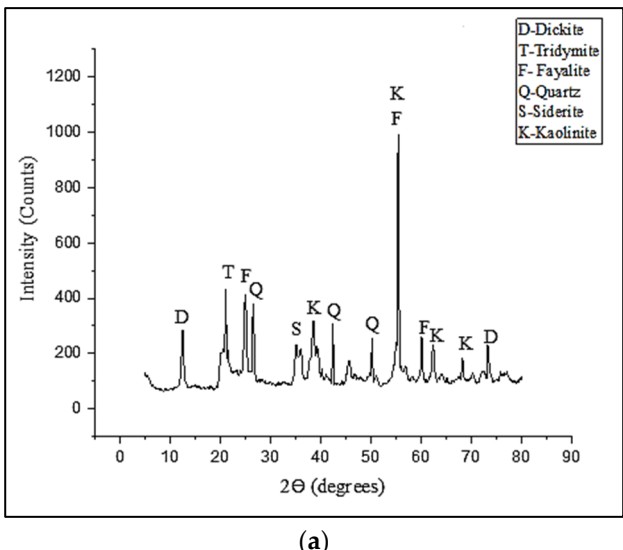
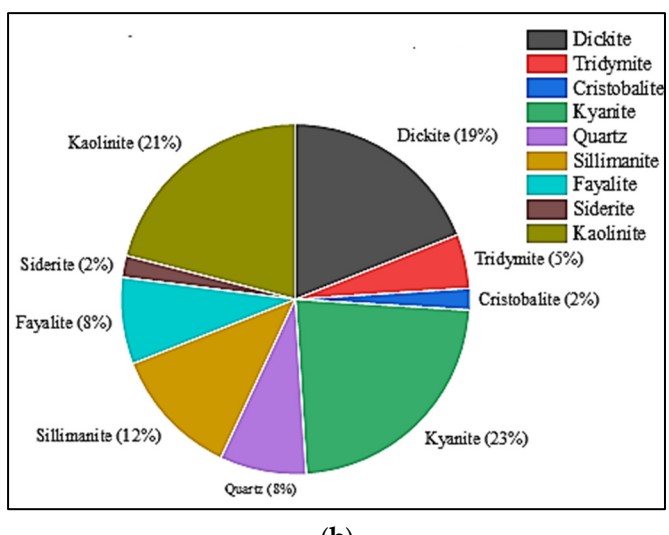

(**a**)                                    (**b**)

**Figure 18.** Characterization of minerals and compounds; (**a**) XRD analysis of Valandhur sample, (**b**) quantification of compounds.

Figure 18a,b shows the minerals and compounds identified through XRD analysis and their quantification respectively for the samples collected from Valandhur location.

The quantification of minerals are as shown in Figure 18b. As per quantification of results, it can be observed that the soil sample consists of relatively higher proportions of Kyanite (23%) followed by kaolinite (21%), dickite (19%), and sillimanite (12%). The quartz (8%), fayalite (8%), tridymite (5%), and siderite (12%) are present at very low proportions. The SEM image as per Figure 17c can be identified as closely packed booklets of dickite. The compounds observed in the XRD analysis are evident from the elements observed by the EDS analysis as shown in Figure 17a,d. The majority of the elements observed are oxygen, silicates, and aluminum followed by iron and carbon. The dickite, quartz, tridymite, kaolinite, siderite minerals can be observed in the peaks shown by XRD analysis (Figure 18a). The elements shown by EDS analysis confirms the presence of aluminum-, iron-, and silicate-based compounds such as siderite, fayalite, tridymite, quartz, and dickite.

Mineralogical analysis of Nagoor samples are shown in Figures 19 and 20. The EDS analysis of Nagoor sample as in Figure 19a,b shows the identified elements weight percentage and atomic weight percentage, respectively, whereas Figure 19c,d show the SEM image and spectrum image of Nagoor sample, respectively.

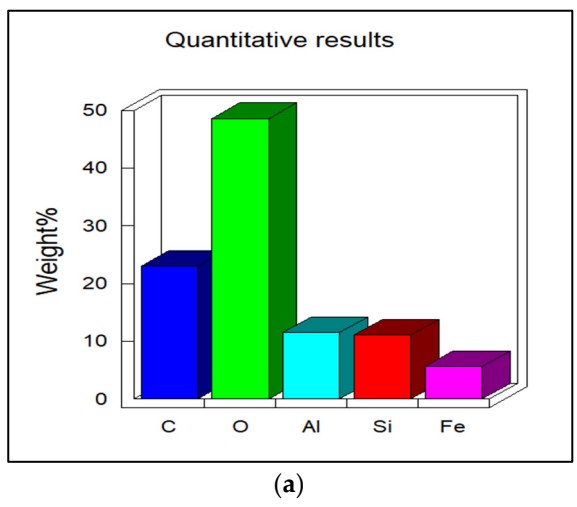
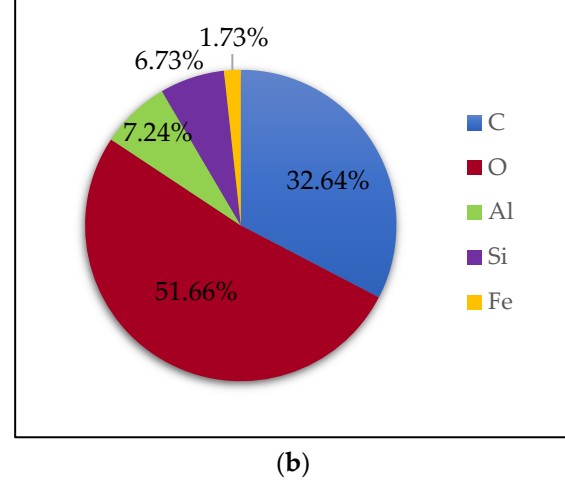

(**a**)                                    (**b**)

**Figure 19.** *Cont.*

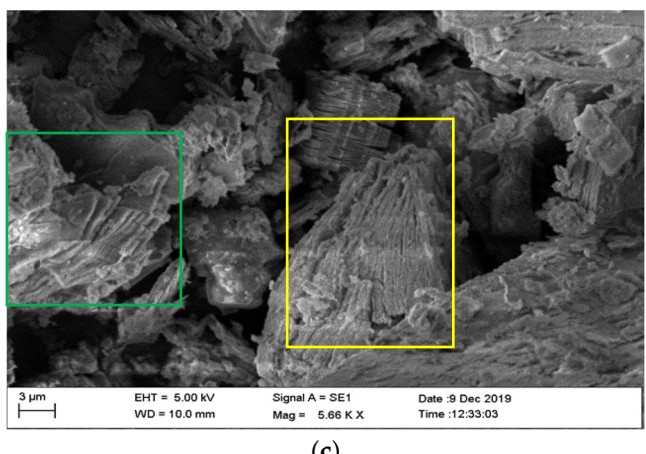

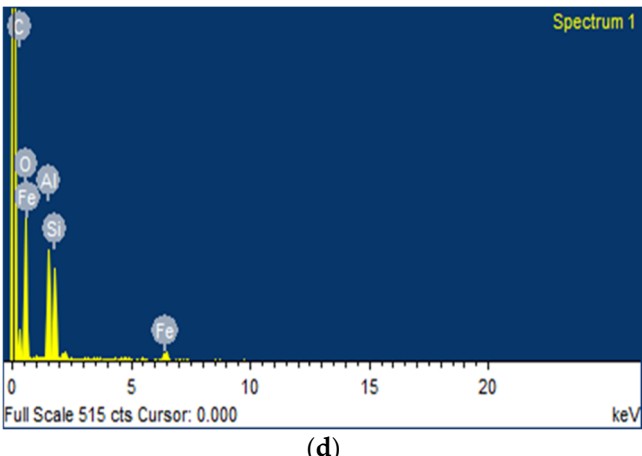

(**c**)  (**d**)

**Figure 19.** EDS analysis of Nagoor sample; (**a**) elements weight (%), (**b**) atomic weight (%), (**c**) SEM image, (**d**) spectrum image.

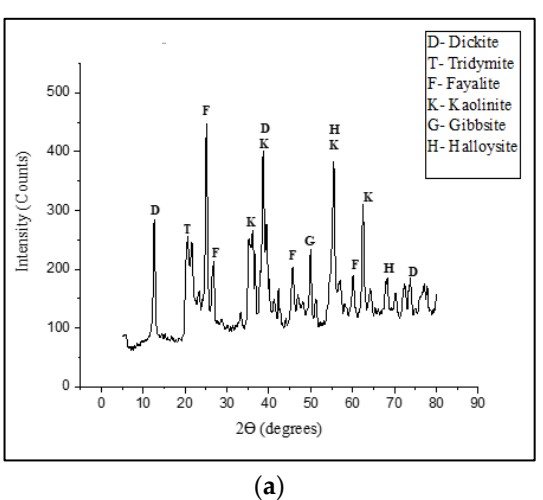

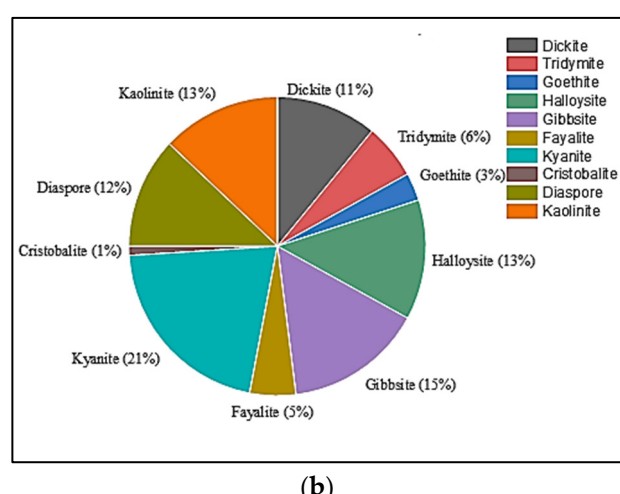

(**a**)  (**b**)

**Figure 20.** Characterization of minerals and compounds; (**a**) XRD analysis of Nagoor sample, (**b**) quantification of compounds.

Figure 20a,b shows the minerals and compounds identified through XRD analysis and their quantification, respectively for the samples collected from Nagoor location.

The figure showing SEM image in Figure 19c can be identified as kaolinite having sheet structured minerals and closely packed dickite. The quantification of minerals are as shown in Figure 20b. As per quantification of results, it can be observed that the soil sample consists of relatively higher proportions of kyanite (21%), kaolinite (13%), halloysite (13%) followed by diaspore (12%), gibbsite (15%), and dickite (11%). The goethite (3%), tridymite (6%), and cristobalite (1%) are present at very low proportions. The compounds observed in the XRD analysis (Figure 20a) are evident from the elements observed by the EDS analysis as shown in the Figure 19a,d. The majority of the elements observed are oxygen, silicates, aluminum, carbon, and iron.

The dickite, fayalite, kaolinite, halloysite, and gibbsite minerals can be observed in the peaks shown by XRD analysis (Figure 20a). The elements shown by EDS analysis confirms the presence of minerals identified.

Mineralogical analysis of Byndoor samples are shown in Figures 21 and 22. The EDS analysis of Byndoor sample as in Figure 21a,b shows the identified elements weight percentage and atomic weight percentage, respectively, whereas Figure 21c,d shows the SEM image and spectrum image of Byndoor sample, respectively.

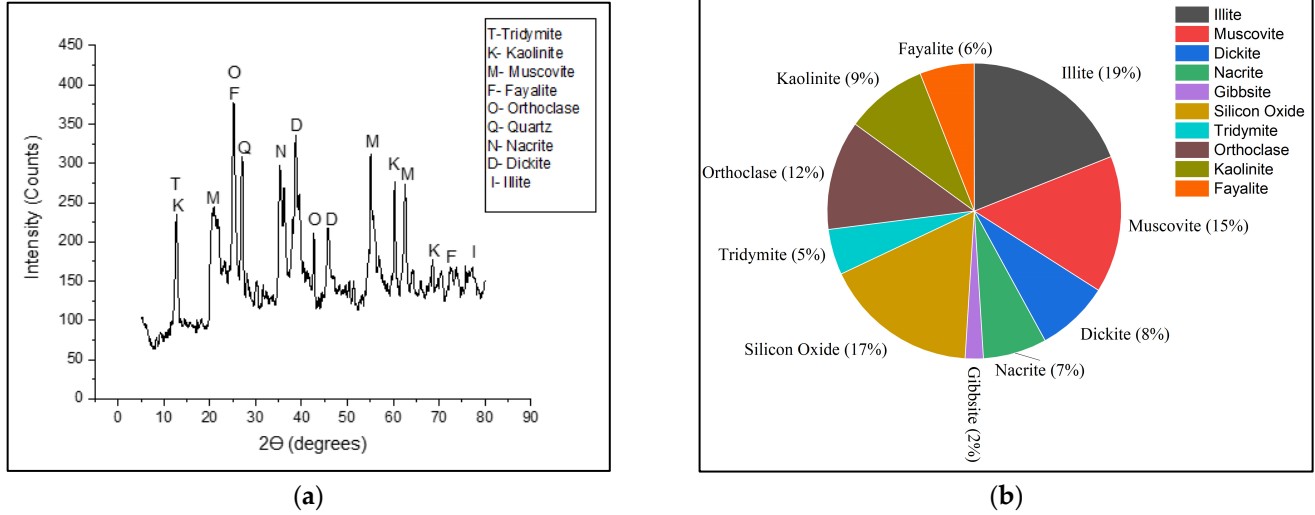

**Figure 21.** EDS Analysis of Byndoor sample; (**a**) elements weight (%), (**b**) atomic weight (%), (**c**) SEM image, (**d**) spectrum image.

**Figure 22.** Characterization of minerals and compounds; (**a**) XRD analysis of Byndoor sample, (**b**) quantification of compounds.

Figure 22a,b shows the minerals and compounds identified through XRD analysis and their quantification, respectively, for the samples collected from Byndoor location.

The SEM image as shown in Figure 21c confirms the presence of sheet minerals with kaolinite and fibrous minerals of illite. The quantification of minerals is as shown in Figure 22b. As per quantification of result, it can be observed that the soil sample consists of relatively higher proportions of illite (19%), muscovite (15%), silicon oxide (17%), orthoclase (12%), and kaolinite (9%). The fayalite (6%), gibbsite (2%), and tridymite (5%) are present at very low proportions. The compounds observed in the XRD analysis (Figure 22a) are evident from the elements observed by the EDS analysis as shown in the Figure 21a,d. The majority of the elements observed are oxygen, silicates, aluminum, iron, potassium, and iron. The kaolinite, illite, muscovite, orthoclase, quartz, dickite, nacrite, and fayalite minerals can be observed in the peaks shown by XRD analysis (Figure 22a). The elements shown by EDS analysis confirms the presence of aluminum, silicate, potassium, aluminum, iron, based compounds such as illite, muscovite, and orthoclase.

Feldspar is a rock-forming silicate mineral which makes up more than 50% of the surface of the earth. In all components of the field, they are found in igneous, metamorphic, and sedimentary rocks. Feldspar minerals have structures, chemical compositions, and bodily properties which are very comparable. The feldspars involves orthoclase ($KAlSi_3O_8$).

The illite type clays are formed under high pH conditions from the weathering of potassium (K) and aluminum (Al) rich rocks. Thus, they are formed through alteration of minerals such as muscovite and feldspar. The Illite clays are the predominant constituent of shales in the soil sample.

Illite contains a diverse category of mica that are 2:1 layer minerals, such as micas and montmorillonites; that is, one octahedral unit is surrounded by two silica tetrahedra units. Illites, however, vary in many ways from mica minerals: one-sixth of the $Si^{4+}$ ions in illites are replaced by $Al^{3+}$ ions, whereas one-fourth of the $Si^{4+}$ ions are replaced in regular micas; and has lesser cation deficiency. A lesser potassium is adsorbed in the interface position in illites. Particles of illite generally are small, less than 1 to 2 microns. This may be attributed to weaker bonding of $K^+$ in the interface positions.

Clay minerals, in particular montmorillonite, exhibit a marked adsorptive capacity; the chemical adsorption by montmorillonite of potassium ions can lead to the formation of illite. They can be commonly found in igneous and metamorphic rocks. Potash feldspar occurs commonly as microcline rather than as an orthoclase [18].

Weathering of feldspars and muscovite leads to the formation of dickite and other kaolin minerals. Different members of the kaolinite subgroup are produced by variations in layer stacking above one other, as well as the position of aluminum ions within the accessible sites in the octahedral sheet. Two unit layers make up a dickite unit cell, while six make up a nacrite unit cell. Both appear to be formed by hydrothermal processes. Dickite is a secondary clay that can be found in sandstone pores and coal seams [19].

## 4. Conclusions

The soil sample collected from various locations represents both primary as well as secondary minerals. The primary minerals observed in most of the locations are quartz, feldspar like orthoclase those that are not altered chemically since the time of formation and deposition. The secondary minerals formed by the decomposition and chemical alteration of primary minerals include kaolinite, dickite, illite, and gibbsite. The samples show the combination of all the minerals in various proportions. The proportion of kaolinite mineral varies from 9 to 27% and is common mineral found at all the locations. Some other minerals such as dickite varies from 8 to 23%, gibbsite varies from 2 to 16%, quartz varies from 4 to 18%, Illite varies from 7 to 25%. The iron compounds such as fayalite vary from 4 to 13%. The common elements present in all site locations as per EDS analysis are Si, Al, Fe, C, and O. The Alevoor sample and Katapadi sample show the presence of chromium (Cr) in addition to common minerals. In addition to the above minerals, the samples from Kumbashi, Manipal, and Kolalgiri also show the presence of niobium (Nb). K is present in

samples of Kumbashi, Hemmadi, Byndoor, and Brahmavar. The presence of K confirms the presence of illite, orthoclase, and other potassium-based compounds.

The soil samples collected from various locations consists of high fines content with high silt content. On saturation, the higher silt content and higher ion hydration or dissociation of sheet minerals affect the strength parameters. The high silt content reduces the plasticity. When water passes through this layer during a monsoon, the silty soils disintegrate and flow like water, frequently washing away the fine soil, forming voids, and occasionally causing substantial settlement and sliding of the upper layers after the application of load. It is necessary to improve the properties of lithomargic clay and the dispersive nature of soil with suitable ground improvement techniques.

**Author Contributions:** Conceptualization, D.N.; methodology, D.N., P.G.S. and U.S.H.N.; validation, D.N. and U.S.H.N.; formal analysis, D.N. and J.B.P.; investigation, D.N., P.G.S. and J.B.P.; resources, D.N., U.S.H.N. and J.B.P.; data curation, D.N., P.G.S. and U.S.H.N.; writing—original draft preparation, D.N.; writing—review and editing, D.N., P.G.S. and U.S.H.N.; visualization, D.N. and P.G.S.; supervision, P.G.S., U.S.H.N. and J.B.P.; project administration, D.N. and P.G.S. All authors have read and agreed to the published version of the manuscript.

**Funding:** This research received no external funding.

**Data Availability Statement:** All data analyzed during this study are included in this published article.

**Conflicts of Interest:** The authors declare that they have no known competing financial interest or personal relationships that could have appeared to influence the work reported in this paper. The authors declare no conflict of interest.

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
