# Peer review of "Mineralogical Characterization of Lithomargic Clay Deposits along the Coastal Belt of Udupi Region of South India"

_jcs, doi:10.3390/jcs7040170_

Round 1

Reviewer 1 Report

The paper does not contain any certain scientific task.

At the same time it looks that it was any attempt to study anthropogenic treatment for soils. But for the latter it is need to use first of all another methods, not local methods (like SEM/EDS), but "bulk" methods, such as chemical analysis and ICP-MS. 

The presented paper does not belong to the direction of the Journal of Composites Sciences, especially to the Section "Polymer Composites" and Special Issue "Advanced Polymeric Composites and Hybrid Materials".

Thus, the material has to be reworked and submitted to another journal.

Author Response

The paper does not contain any certain scientific task. At the same time it looks that it was any attempt to study anthropogenic treatment for soils. But for the latter it is need to use first of all another methods, not local methods (like SEM/EDS), but "bulk" methods, such as chemical analysis and ICP-MS. The presented paper does not belong to the direction of the Journal of Composites Sciences, especially to the Section "Polymer Composites" and Special Issue "Advanced Polymeric Composites and Hybrid Materials". Thus, the material has to be reworked and submitted to another journal.

Response: The constituent rocks and minerals have notably dissimilar chemical or physical properties and are merged by weathering and hydrothermal alteration on variable temperature and pressure to create a material like lithomargic clay with properties unlike the individual elements which substantiate the definition of composite material. Within the finished structure, the individual elements remain separate and distinct, distinguishing composites from individual elements which can be simulated to the behaviour of composite material.

As the minerals present in the soil are responsible for the physical and chemical behaviour of soil, it is very important to know the minerals present and its variation along the study region. By knowing the mineral characteristics and its behaviour, one can decide on the suitable ground improvement techniques to be applied on soil to overcome the shortcomings in soil. In the present study, characterization and mineralogical properties are evaluated to identify the presence of minerals and compounds for the various soil samples collected along the coastal belt of Udupi regions using X-Ray diffraction (XRD), Energy dispersive X-ray Spectroscopy (EDS) and Scanning Electron microscopy (SEM) analysis to discuss and find the correlation of mineralogy with geotechnical parameters and suggest the suitable ground improvement techniques to be applied on soil. The properties of lithomargic clays can be improved by blending with natural fibres as cited in [8,9] or improved by blending with granulated blast furnace slag as cited in [11,14,16] or can be improved by blending with quarry dust and cement as cited in [3,15].

Reviewer 2 Report

The manuscript list mineralogical and element data of the lithomargic clay samples from coastal belt of Udupi region, South India. Large data-set are show and the characteristic of mineral content are described. The manuscript need a intermediate changes.

1)     Stratigraphy is very essential for readers to understand the forming process of geological units and estimate their potential engineering feature. I suggest authors depict the classical section(s) studied and show it in figure in this manuscript. Readers can comprehend the relationship in lateritic crust, lithomargic clay, ferricretic layer, ferruginous zone, limonitic, transition and saprolitic zones.

2)     I do not agree that “The increase in strength was discovered by SEM and XRD research”in Lines 67-68 and “The increase in strength was determined by SEM and XRD examinations” in Lines 91-92. The strength of residual soil cannot be discovered on the basis of research on SEM and XRD. Images of SEM can show microstructure and types of minerals, and XRD can be used to analyze types of minerals. Strength is a kind of mechanical property. There are some other special methods (or instruments) to measure it.

3)     I did not understand the necessity of the sentences in Lines 52-101 and logic between them and this manuscript. They did not show the significances of the manuscript, yet the problems need to be solved. The introduction needs a further improvement to make the purpose of the manuscript clearer.

4)     It is necessary for authors to explain why they select 75μ as threshold for samples to measure and if mineral assemblages are same between bulk samples and theirs segments that can pass through 75μ. How could they make sure the fidelity of measurement? It is the similar question for authors to pick the 425μ as threshold for samples in order to measure using X-Ray Diffraction.

5)     Authors list all the data measured for samples, but they did not compile and discuss them for their theme.

6)     Topic of the manuscript is about the mineralogical characterization. Authors only concern the content of the minerals, it is not all-sided.

7)     Please show the meaning of the GBFS in Line 66, CSRE in Line 95, when they occur firstly in manuscript.

Author Response

  1. Stratigraphy is very essential for readers to understand the forming process of geological units and estimate their potential engineering feature. I suggest authors depict the classical section(s) studied and show it in figure in this manuscript. Readers can comprehend the relationship in lateritic crust, lithomargic clay, ferricretic layer, ferruginous zone, limonitic, transition and saprolitic zones.

Response:  A stratigraphy showing the different soil layers are included as follows with citation [1].

Kindly refer to the revised manuscriot for Fig.1: Soil stratification (a) Laterization on soil made up of gneissic granites (b) showing different zones of the laterite profile [1]

  1. I do not agree that “The increase in strength was discovered by SEM and XRD research “in Lines 67-68 and “The increase in strength was determined by SEM and XRD examinations” in Lines 91-92. The strength of residual soil cannot be discovered on the basis of research on SEM and XRD. Images of SEM can show microstructure and types of minerals, and XRD can be used to analyze types of minerals. Strength is a kind of mechanical property. There are some other special methods (or instruments) to measure it.

Response:

The above statements of increase in the strength was part of our literature survey. The citation is also added to the above mentioned statement at the end of the literature. The citation number [10] and [13] justify the formation of compounds responsible for strength calcium silicate hydrates (C-S-H), calcium aluminate silicate hydrates (C-A-S-H), calcium silicate hydroxide hydrate (CSHH). The authors conducted the geotechnical tests like to confirm the increase in strength.

  1. I did not understand the necessity of the sentences in Lines 52-101 and logic between them and this manuscript. They did not show the significances of the manuscript, yet the problems need to be solved. The introduction needs a further improvement to make the purpose of the manuscript clearer.

Response: A sentence is added in the introduction about the importance of characterisation and mineralogy a step towards solving the weak problems in lithomargic clay.  “As the minerals present in the soil are responsible for the physical and chemical behaviour of soil, it is very important to know the minerals present and its variation along the study region. By knowing the mineral characteristics and its behaviour, one can decide on the suitable ground improvement techniques to be applied on soil to overcome the shortcomings in soil. As very less number of literatures are available on lithomargic clay, the literatures on mineralogy and literatures on correlation of mineralogy with geotechnical parameters are included in the introduction. Now added citation to justify the compounds responsible for increase in the strength of lithomargic clay.

  1. It is necessary for authors to explain why they select 75μ as threshold for samples to measure and if mineral assemblages are same between bulk samples and theirs segments that can pass through 75μ. How could they make sure the fidelity of measurement? It is the similar question for authors to pick the 425μ as threshold for samples in order to measure using X-Ray Diffraction.

Response: As the present paper is part of geotechnical and geological correlation study and IS code specifies that for some of geotechnical test the samples must be less than 75 microns and some tests it must be less than 425 microns. The elements and minerals are very small in size and lithomargic clay is having high content of sit and clay having size less than 75 microns. Also It is to have just enough particles in the x-ray interaction volume of XRD set up in order to have a nearly homogenous distribution of crystallites in angular space. Otherwise we may suffer from orientational artifacts, which will lead to variations in peak heights compared to the ideal case. Errors in peak heights will lead to errors in phase composition derived from XRD.

  1. Authors list all the data measured for samples, but they did not compile and discuss them for their theme.

Response: In Results and discussion, under section 3, all the elements identified from Energy dispersive X-ray Spectroscopy (EDS), compounds identified from the Scanning Electron microscopy (SEM) and minerals or compounds identified using the results obtained from X-Ray diffraction (XRD) tests are discussed in detail from page no. 6 to page number 24 with each sample collected from different locations. On page No. 24 also discussed about the formation of different secondary minerals identified from the study. In conclusion, concluded on the compiled results.

  1. Topic of the manuscript is about the mineralogical characterization. Authors only concern the content of the minerals, it is not all-sided.

Response: As the title/ theme is mineral characterization, for all the samples collected from different locations the variations of minerals identified through XRD test results, their chemical formula with correlation of elements observed through EDS analysis to justify the presence of compounds with the help of SEM image.

  1. Please show the meaning of the GBFS in Line 66, CSRE in Line 95, when they occur firstly in manuscript

Response:

The meaning is included in the manuscript. GBFS is Granulated blast furnace slag and CSRE is Cement stabilised rammed earth.

Reviewer 3 Report

Overall view:

Minor comment

Comment:

Overall, this paper provides a comprehensive characterization and mineralogical study of lithomargic clay found along the coastal belt of the Udupi district in South India. The study used a variety of techniques including X-Ray diffraction (XRD), Energy dispersive X-ray Spectroscopy (EDS) and Scanning Electron microscopy (SEM) to identify the presence of minerals and compounds in the soil samples.

However, there are a few areas where I believe the paper could be improved:

1.      References are not sufficient and more related works should be described in the Introduction. Below are some examples:

a.      C Sekhar, D., & Nayak, S. (2019). SEM and XRD investigations on lithomargic clay stabilized using granulated blast furnace slag and cement. International Journal of Geotechnical Engineering, 13(6), 615-629.

b.      Amulya, S., Ravi Shankar, A. U., & Praveen, M. (2020). Stabilisation of lithomargic clay using alkali activated fly ash and ground granulated blast furnace slag. International Journal of Pavement Engineering, 21(9), 1114-1121.

c.      Sarvade, P. G., & Nayak, S. (2011). Microfabric and mineralogical studies using sem and xrd on the lithomargic clay stabilized with cement and quarry dust. International Journal of Earth Sciences and Engineering, 4(5), 266-273.

2.      The labels in Figure 5b are not clear. Please replace it with high resolution one.

3.      The caption of Fig. 6, 8, 10, 12, 16, 18, and 20 are the same. Add the name of the sample to captions (Alevoor, Kolalgiri, Brahmavar).

4.      The description of results and sample properties should be quantitative in the Conclusion.

5.      The format of referred captions should be according to the journal's format. So, “Fig.” should be replaced with “Figure” throughout the manuscript.

Author Response

Overall, this paper provides a comprehensive characterization and mineralogical study of lithomargic clay found along the coastal belt of the Udupi district in South India. The study used a variety of techniques including X-Ray diffraction (XRD), Energy dispersive X-ray Spectroscopy (EDS) and Scanning Electron microscopy (SEM) to identify the presence of minerals and compounds in the soil samples.

However, there are a few areas where I believe the paper could be improved:

  1. References are not sufficient and more related works should be described in the Introduction. Below are some examples:
  2. C Sekhar, D., & Nayak, S. (2019). SEM and XRD investigations on lithomargic clay stabilized using granulated blast furnace slag and cement. International Journal of Geotechnical Engineering, 13(6), 615-629.
  3. Amulya, S., Ravi Shankar, A. U., & Praveen, M. (2020). Stabilisation of lithomargic clay using alkali activated fly ash and ground granulated blast furnace slag. International Journal of Pavement Engineering, 21(9), 1114-1121.
  4. Sarvade, P. G., & Nayak, S. (2011). Microfabric and mineralogical studies using sem and xrd on the lithomargic clay stabilized with cement and quarry dust. International Journal of Earth Sciences and Engineering, 4(5), 266-273.

Response: The additional references have cited in the introduction and included in the references.

  1. The labels in Figure 5b are not clear. Please replace it with high resolution one.

Response: The Figure 5b is replaced with high resolution image.

  1. The caption of Fig. 6, 8, 10, 12, 16, 18, and 20 are the same. Add the name of the sample to captions (Alevoor, Kolalgiri, Brahmavar).

Response: All the above mentioned figure captions are modified as suggested.

  1. The description of results and sample properties should be quantitative in the Conclusion.

Response: Conclusion is updated with quantification of minerals.

  1. The format of referred captions should be according to the journal's format. So, “Fig.” should be replaced with “Figure” throughout the manuscript.

Response:  The Fig is replaced with Figure throughout the manuscript as suggested.

Round 2

Reviewer 1 Report

The paper does not correspond to the journal "Journal of Composites Science", the section "Polymer Composites" and its special issue "Advanced Polymeric Composites and Hybrid Materials".

Please see the information to the journal special issue:

This Special Issue focuses on the investigation of revolutionary new composite material formulations, special treatments, recyclability, intelligent features, engineering phenomena, and novel manufacturing concepts that are ushering in new composite material trends in the aerospace, automotive, and all engineering sectors.

In this Special Issue of Advanced Polymeric Composites and Hybrid Materials, we would like to invite authors to submit original papers and reviews on the topic to disseminate findings from studies conducted on materials, including ceramics, glasses, polymers (plastics), composites, semiconductors, magnetic materials, biological and biomimetic materials, silica, carbon, and dot materials. Both academic and industry researchers are encouraged to submit their findings and new developments in this area for publication. Research works that focus on progressive materials and technologies, new characterization techniques to study the relationship between microstructure and structural properties, and also physical and numerical simulation studies are especially encouraged.

The content of the reviewed paper “Mineralogical Characterization of Lithomargic Clay Deposits from Coastal belt of Udupi region of South India” does not input into the frames of the issue and for the journal direction as a whole.

Thus, the paper “Mineralogical Characterization of Lithomargic Clay Deposits from Coastal belt of Udupi region of South Indiahas be submitted to another appropriate journal.

Also, following to the very low scientific level mineralogical data presentation, analysis and interpretation, the paper has to be reworked and resubmitted for new reviewing to any geological or mineralogical journal.

Reviewer 3 Report

The revised version is acceptable for publication. However, the format of labels for figures should be corrected.